Construction and validation of a PANoptosis-related lncRNA signature for predicting prognosis and targeted drug response in thyroid cancer

Li Ruowen 1 2
Zhao Mingjian 1 2
Sun Min 1 2
Miao Chengxu 1 2
Lu Jinghui 1 jinghuilu@email.sdu.edu.cn
1 Qilu Hospital, Cheeloo College of Medicine, Shandong University , Jinan, Shandong , China
2 School of Medicine, Cheeloo College of Medicine, Shandong University , Jinan, Shandong , China
Gould Gwyn
Electronic publication date: 2023 Sep 1
Publication date: 2023
Volume: 11
Electronic Location ID: e15884
Received 2023 Apr 12; Accepted 2023 Jul 20
Copyright: © 2023 Li et al.
Copyright year: 2023
Copyright holder: Li et al.
License: This is an open access article distributed under the terms of the Creative Commons Attribution License, which permits unrestricted use, distribution, reproduction and adaptation in any medium and for any purpose provided that it is properly attributed. For attribution, the original author(s), title, publication source (PeerJ) and either DOI or URL of the article must be cited.
License URL: https://creativecommons.org/licenses/by/4.0/

Keywords: PANoptosis, Thyroid cancer (TC), Long non-coding RNA (lncRNA), Prognosis, Tumor immune microenvironment, Proliferation, Migration, Drug response

Funding: Natural Science Foundation of Shandong Province ZR2021LZL003 This study was supported by grants from the Natural Science Foundation of Shandong Province (No. ZR2021LZL003). The funders had no role in study design, data collection and analysis, decision to publish, or preparation of the manuscript.

==============================
Thyroid cancer (TC) is the most prevalent malignancy of the endocrine system. PANoptosis, a newly discovered cell death pathway, is of interest in tumor research. However, the relationship between PANoptosis-related lncRNAs (PRlncRNAs) and TC remains unclear. The study aimed to develop a prognostic model based on PRlncRNAs in TC. Gene expression data of PANoptosis-associated genes and clinical information on TC from The Cancer Genome Atlas (TCGA) database were analyzed by Pearson correlation analysis, univariate/multivariate Cox analysis, and Lasso Cox regression analysis. A PRlncRNA signature was constructed and used to develop a nomogram to predict overall survival (OS). We further explored the correlation between the risk score and tumor immune microenvironment, immune checkpoints, and drug sensitivity. Moreover, we verified the expression and biological function of lncRNAs in TC cell lines. Finally, seven PRlncRNAs were used to construct a prognostic model for predicting the OS of TC patients. We found that the risk score was associated with the tumor microenvironment (TME) and the expression of critical immune checkpoints. In addition, we screened for drugs that high- or low-risk TC groups might be sensitive to. Quantitative real-time polymerase chain reaction (qRT-PCR) results showed differential expression of four PRlncRNAs (GAPLINC, IDI2-AS1, LINC02154, and RBPMS-AS1) between tumor and normal tissues. Besides, a GEO database (GSE33630) was used to verify the expression differences of PRLncRNAs in THCA tissues and normal tissues. Finally, RBPMS-AS1 was found to inhibit the proliferation and migration of TC cells. In conclusion, we developed a PANoptosis-related lncRNA prognostic risk model that offers a comprehensive understanding of TME status in patients with TC and establishes a foundation for the choice of sensitive medications and immunotherapy.

Introduction

Thyroid cancer (TC) is the most prevalent malignancy of the endocrine system, and its incidence has rapidly increased worldwide in recent years (Lim et al., 2017). Owing to advances in surgery and radioiodine therapy, most patients with TC usually have a relatively good prognosis; however, patients with advanced TC, especially iodine-refractory TC, still lack effective treatment strategies, with a 10-year survival rate of only 10% (Durante et al., 2006). Despite the existence of some biomarkers of prognosis for TC, accurate prediction of progression in patients with thyroid cancer remains difficult. Therefore, supplementing and improving traditional risk stratification methods for TC patients remains an urgent problem.

With the development of second-generation sequencing, long non-coding RNA (lncRNA) has been studied widely. lncRNAs are a category of noncoding RNAs containing over 200 nucleotides that are primarily transcribed by RNA enzyme II and lack an evident open reading frame (Li et al., 2016). These lncRNAs primarily regulate cellular function, including cancer development and progression (Rao, Rajkumar & Mani, 2017). Emerging studies have found that lncRNA expression has significant tissue- and disease-specificity, making it a potential biomarker for early tumor development (Tornesello et al., 2020). Guo et al. (2022) revealed that a model composed of seven prognostic lncRNAs of papillary thyroid cancer (PTC). Qin et al. (2022) validated a signature supported by four Ferroptosis-related lncRNAs as predictive markers for TC. Numerous studies have shown that lncRNAs influence TC prognosis, which means that we could develop a more accurate TC biomarker based on lncRNAs.

Programmed cell death (PCD) is a pathological form of cell death induced by an intrinsic program, the most well-defined of which are apoptosis, pyroptosis, and necroptosis. Apoptosis is a widely studied cell death mechanism activated by the initiator cysteine-8/10 and the downstream effectors cysteine-3 and -7 (Fleisher, 1997). When apoptosis is ineffective, pyroptosis or necroptosis becomes an alternative PCD pathway for eliminating cancer cells (Bertheloot, Latz & Franklin, 2021). Pyroptosis is executed by members of the gasdermin family, whereas necroptosis is mediated through receptor-interacting protein kinase 3 (RIPK3)-dependent mixed lineage kinase-like (MLKL) oligomerization (Frank & Vince, 2019). In 2019, Malireddi, Kesavardhana & Kanneganti (2019) observed for the first time that apoptosis, pyroptosis, and necrosis are not independent cell death pathways as commonly recognized but have potential interactions and named the combined inflammatory cell death “PANoptosis”. Most importantly, PANoptosis cannot be explained by any of these three PCD pathways alone. Recently, accumulating evidence has shown that PANoptosis plays a critical role in cancer (Karki et al., 2020, 2021). The goals of the present study were to develop a clinical prediction model based on PANoptosis-related lncRNAs, which may cast light on designing and improving therapeutic strategies.

Methods

Functional analysis of PANoptosis-related genes

Based on previous literature and databases, we identified 35 PANoptosis-related genes (PRGs) (Table S1). The selected PRGs were characterized by Gene ontology (GO) and Kyoto Encyclopedia of Genes and Genomes (KEGG) enrichment utilizing the “clusterprofiler” package of R software (version 4.0.5; R Core Team, 2021). The results were visualized and plotted with the help of “enrichplot” and “ggplot2” packages.

Identification of PANoptosis-related lncRNAs

Gene expression data of 571 TC samples (512 tumor and 59 normal) and matching clinical data of 507 TC patients (Patients with incomplete tracking data were not included) were acquired from The Cancer Genome Atlas (TCGA) databank (https://portal.gdc.cancer.gov/). The expression data is available at https://www.jianguoyun.com/p/DeqNJTAQ9pbeCxjIxY0FIAA, and clinical data are listed in Table S2. To identify PRlncRNAs, Pearson correlation was used to determine the correlation between lncRNAs and PRGs in the present study. Correlation coefficient |R2| > 0.4 and p < 0.001 were used as inclusion criteria for PRlncRNAs. The differentially expressed LncRNAs in TC were obtained using the “limma” package.

Establishment of the prognostic model

We randomly distributed all TC patients in the TCGA dataset into the training set (n = 256) and the test set (n = 256) in a 1:1 ratio, and the training set was used for model construction. Data from the test set and entire set were utilized to estimate the performance of the prognostic model. The “Survival” package was applied to conduct univariate/multivariate Cox analysis for PRlncRNAs with p-value ≤ 0.05. Subsequently, we determine the optimal prediction model by means of the least absolute choice operator (LASSO) (Simon et al., 2011). The risk scores of individual patients were computed by incorporating the expression level of lncRNAs in the model and the regression coefficient in the multivariate analysis. The formula is as follows: Σcoefficient of (PRlncRNAi) × expression of (PRlncRNAi).

Validation of model

The samples were categorized as high- or low-risk according to their average risk scores. Kaplan–Meier method was used for survival analysis of high-risk and low-risk groups. The “pheatmap” package was used to map the risk profile, survival status, and lncRNA expression heat map of the groups. The “survivalROC” and “timeROC” packages were used to plot the Receiver Operating Characteristic (ROC) curve and the Area Under the Curve (AUC) to determine the level of accuracy of the signature. Univariate and multivariate Cox regression analyses were performed to verify whether the predictive signature was an independent prognostic variable for Overall Survival (OS). Using the “rms” R package (Iasonos et al., 2008), we combined the OS of TC patients with age, gender, and tumor stage to create a nomogram. A calibration curve was used to test the agreement between the actual results and model predictions according to the Hosmer–Lemeshow test. To compare the accuracy, we calculated the consistency index (c-index) and restricted mean survival (RMS) of the published models (Guo et al., 2022; Qin et al., 2022; Ding et al., 2022; Liu et al., 2022; Shan et al., 2022) and the model we developed.

Gene enrichment analysis

To explore the molecular and biological differences between these two groups, the KEGG and HALLMARK gene sets referenced by the molecular signature database (https://www.gsea-msigdb.org/gsea/msigdb/index.jsp) were analyzed by the “clusterProfiler” package in the TCGA cohort (p < 0.05, FDR < 0.25). Single-sample GSEA (ssGSEA) was conducted for significant gene sets using the “GSVA” package (Hänzelmann, Castelo & Guinney, 2013).

Estimation of tumor-infiltrating immune cells

THCA immune cell infiltration assessment data were were firstly estimated by the Tumor Immune Estimation Resource (http://timer.cistrome.org/). Based on the TCGA database expression data, we applied the TIMER (Li et al., 2017), CIBERSORT (Newman et al., 2015), CIBERSORT-ABS, QUANTISEQ, MCPcounter (Dienstmann et al., 2019), xCELL (Aran, Hu & Butte, 2017), and EPIC (Racle & Gfeller, 2020) algorithms to obtain the cell composition for the TCGA-THCA dataset samples by R software (Table S3). This enabled comparison of immune cell subpopulations in the groups. ssGSEA was also performed using the “GSVA” packages in R to compare the immune-cell infiltration in the groups. ImmuneScore, StromalScore, and ESTIMATEScore (ImmuneScore + StromalScore) were quantified for each patient using the “limma” and “estimate” packages in R. Finally, the “ggpubr” package in R was used to map immune checkpoints for risk groups (Charoentong et al., 2017).

Clusters based on prognostic PRlncRNAs

Given the PRlncRNA expression associated with the risk signature, we obtained two subtypes using the “ConsensusClusterPlus” package in R. We then evaluated the differences in immune cell infiltration and immune scores between the two subtypes.

Drug sensitivity prediction

The accuracy of the signature in forecasting the response to common antitumor drugs on TC was also evaluated. To determine the therapeutic response, the half inhibitory concentration (IC50) of each sample was estimated with the help of the “pRRophetic” package (Geeleher, Cox & Huang, 2014). Lower IC50 values indicate higher drug sensitivity. The Wilcoxon test was used to calculate the p-values of differences in drug sensitivity.

Quantitative PCR

Ten paired tumor and para-tumor TC tissue samples were acquired from Qilu Hospital of Shandong University. Our study was approved by the Research Ethics Committee of Qilu Hospital of Shandong University (Approval Number: KYLL-2017(KS)-248), and all patients signed a written informed consent form. RNA was isolated from tissues with the total RNA extraction reagent TRIzol (DP424; TIANGEN Biotech, Beijing, China). The obtained RNA was then reverse-transcribed to cDNA using HiScript II Reverse Transcriptase (Vazyme, Nanjing, China). Quantitative PCR assays were performed using a Bio-Rad C1000 Thermal Cycler with SYBR Green Master Mix (Q711; Vazyme Biotech, Nanjing, China), and expression levels were calculated using the 2−ΔΔCt method. β-actin was used as a standard control. The primers designed for qRT-PCR were obtained from Sangon Biotech (Shanghai, China). And the sequences are presented in Table S4.

Cell culture and transfection

The human thyroid cell lines TPC-1 (RRID: CVCL_6298) was obtained from the Stem Cell Bank, Chinese Academy of Sciences (Shanghai, China). The PTC cell line BHP10-3 (RRID: CVCL_6278) were supplied by Qilu Hospital of Shandong University (Zhang et al., 2019). TPC-1 and BHP10-3 cells were cultured in Roswell Park Memorial Institute (RPMI 1640) complete medium (Shanghai Institute of Biological Sciences, Shanghai, China) at 37 °C in an incubator with a constant temperature and 5% CO2. The cells were dispersed in six Well Cell Culture Plates and transfected the following day. The overexpression plasmid was obtained from MiaoLing Plasmid Platform (Wuhan, China). When the cell density reached approximately 70%, 2,000 ng of negative control or plasmid was transfected into cells using Lipofectamine™ 2000 Transfection Reagent (11668019; Invitrogen, Waltham, MA, USA). After 6 h, the medium was replaced with 2 mL of complete medium. After 24 h of transfection, cells were collected for subsequent experiments.

Proliferation and migration assay

After 24 h of transfection, the cells were evenly distributed in 96-well plates, five parallel replicate wells were set for each group, and 10 μL of CCK-8 reagent was added on days 1, 2, and 3. The absorbance was measured at 450 and 630 nm, which represents the cell growth rate. Thyroid cancer cells (5 × 104) transfected with RBPMS-AS1 overexpression plasmid or negative control were introduced to the upper chamber of Transwell plates in 200 μL serum-free culture medium, and 800 μL medium comprising 10% fetal bovine serum was introduced into the lower chamber as an inducer. After 24 h, the migrated cells were washed with PBS, fixed in methanol, stained with 1% crystal violet, and non-migrating cells in the upper chamber were removed with a cotton swab. At least four randomly selected fields were observed and counted under a microscope (Olympus, Tokyo, Japan).

Wound healing assay

The cells were incubated in 6-well plates. When the cells were cultured to 80% confluence, a 200 μL pipette tip was used to create a scratch. The cells were then washed with PBS and placed in serum-deficient medium for an additional 24 h. The wound healing area at 0 and 24 h was recorded using a microscope (Olympus, Tokyo, Japan).

Results

Functional enrichment analysis of PANoptosis-related genes

From the previous literature, we obtained 35 PRGs. Based on GO analysis, PRGs primarily participate in biological processes (BP), such as the positive regulation of cytokine production, positive regulation of response to external stimulus, and response to virus. GO analysis of cellular components (CC) revealed that PRGs were mainly positioned in the membrane raft, membrane microdomain, and inflammasome complex. According to molecular function (MF), PRGs mostly carried out the cytokine receptor binding function (Figs. 1A, 1B). KEGG enrichment analysis showed that the selected PRGs were mainly involved in signaling pathways including the NOD-like receptor signaling pathway, Influenza A-mediated pathways, and necroptosis (Figs. 1C, 1D).

Figure 1 Functional enrichment analysis of PANoptosis-related genes.

(A, B) Gene ontology analysis. (C, D) KEGG pathway enrichment analysis.

Construction of the PANoptosis-related lncRNAs predictive signature in TC

Using Pearson correlation analysis, we recognized 2,192 PANoptosis-related lncRNAs (PRlncRNAs). The Sankey Diagram (Yue et al., 2022) shows the relationship between PRGs and lncRNAs (Fig. S1A). We then obtained 819 differentially expressed PRlncRNAs between the normal and tumor samples (log2|FC| > 1, FDR < 0.05). The volcano map is shown in Fig. 2A. Univariate Cox analysis were performed to investigate the link between lncRNA expression and survival. The analysis indicated that 24 lncRNAs were significantly correlated with the OS of TC patients (Fig. 2B). Notably, 21 PRlncRNAs were “risk” genes, while only LINC0115, DPP4-DT, and RBPMS-AS1 could be considered “protective” genes. The heatmap visualizes PRlncRNA expression in normal and tumor samples (Fig. 2C). LASSO Cox regression analysis was conducted to select and validate prognostic lncRNAs (Figs. 2D, 2E). Finally, the multivariate Cox regression analysis selected seven PRlncRNAs for the construction of the risk signature. A PANoptosis-related prognostic lncRNA signature for TC was created by merging the regression factors and expressive values of seven lncRNAs: IDI2-AS1, LINC02154, DPP4-DT, RBPMS-AS1, C5orf34-AS1, LINC01705, and GAPLINC (Fig. 1F). The expression of the seven lncRNAs was closely correlated with PRGs (Fig. 2F).

Figure 2 Construction of the PRlncRNA predictive signature in thyroid cancer (TC).

(A) Volcano plot of differentially expressed PRlncRNAs in TC. (B) Forest plots of the univariate Cox regression analysis for 23 PRlncRNAs. (C) Heatmap shows the expression of the prognostic PRlncRNAs in normal and tumor. (D, E) LASSO Cox regression analyzing and multivariate Cox regression analyzing. (F) The correlations between the PANoptosis-related genes and the seven prognostic PRlncRNAs.

Evaluating the predictive capacity of the risk signature

To examine the efficacy of the model, the patients in the public database were assigned to training and validation cohorts. The basic information of the patients in both sets is listed in Table 1. Based on the risk score, the patients in each set were classified into high-risk and low-risk groups. Patients in the high-risk group had poor survival according to the Kaplan–Meier curve analysis (p < 0.001) (Fig. 3A). Figures 4B and 4C show the survival status, distribution of patients, and the mortality rate increased with risk scores. Both testing and the entire set suggested a poor prognosis in high-risk patients (Figs. 3F–3H, 3K–3M). Furthermore, ROC was used to validate the risk signature. The AUC values of this risk signature were 1.000, 0.986, and 0.882 for 1, 3, and 5 years respectively, suggesting good predictive power (Fig. 3D). In addition, the ROC analysis results had high accuracy for both the testing set and the entire set (Figs. 3I, 3M). Figures 3E, 3J and 3O depict the heat maps of the expression levels of the seven PRlncRNAs in all sets.

Table 1 The basic information of TCGA-THCA.

Variables	Type	Total	Test	Train	P value	
Age	<=65	433 (85.91%)	225 (89.29%)	208 (82.54%)	0.0405	
	>65	71 (14.09%)	27 (10.71%)	44 (17.46%)		
Gender	Female	369 (73.21%)	184 (73.02%)	185 (73.41%)	1	
	Male	135 (26.79%)	68 (26.98%)	67 (26.59%)		
Stage	Stage I	283 (56.15%)	155 (61.51%)	128 (50.79%)	0.1161	
	Stage II	52 (10.32%)	23 (9.13%)	29 (11.51%)		
	Stage III	112 (22.22%)	49 (19.44%)	63 (25%)		
	Stage IV	55 (10.91%)	24 (9.52%)	31 (12.3%)		
	Unknow	2 (0.4%)	1 (0.4%)	1 (0.4%)		
T	T1	143 (28.37%)	70 (27.78%)	73 (28.97%)	0.2742	
	T2	166 (32.94%)	85 (33.73%)	81 (32.14%)		
	T3	170 (33.73%)	88 (34.92%)	82 (32.54%)		
	T4	23 (4.56%)	7 (2.78%)	16 (6.35%)		
	Unknow	2 (0.4%)	2 (0.79%)	0 (0%)		
M	M0	282 (55.95%)	145 (57.54%)	137 (54.37%)	0.4656	
	M1	9 (1.79%)	3 (1.19%)	6 (2.38%)		
	Unknow	213 (42.26%)	104 (41.27%)	109 (43.25%)		
N	N0	229 (45.44%)	118 (46.83%)	111 (44.05%)	0.5106	
	N1	225 (44.64%)	108 (42.86%)	117 (46.43%)		
	Unknow	50 (9.92%)	26 (10.32%)	24 (9.52%)		

Figure 3 Prognostic evaluation of risk signature.

K-M survival curves, survival time and survival status, 1-, 3-, and 5-year receiver operating curves, heat maps of seven lncRNA expressions in the training (A–E), testing (F–J), and the entire sets (K–O), respectively.

Figure 4 K-M survival analysis of patients with different clinicopathological features.

(A) Patients with MALE. (B) Patients with FEMALE. (C) Patients with age ≤65. (D) Patients with T1–T2. (E) Patients with T3–T4. (F) Patients with N0. (G) Patients with N1. (H) Patients with M0. (I) Patients in stages III–IV.

Association of clinicopathological factors with PRlncRNAs risk signature

To verify the predictive effect on patients with different clinicopathologic variables, patients with TC were stratified in terms of age, gender, and clinical stage. Patients in the high-risk group had notably shorter OS in all subgroups of age (≤65 years), gender (female or male), TNM stage (T1-2 or T3-4, N0 or N1, M0), and clinical stage (stage–III–IV) (Fig. 4). Moreover, the model demonstrated the highest sensitivity and specificity compared to other clinical variables such as age, gender, grade, and stage in all sets (Figs. 5A–5C). To better assess the independent forecasting capacity of the signature, the Univariate/Multivariate Cox regression analyses were conducted on risk scores and patient clinical data. Univariate Cox regression analysis revealed that age, stage, and risk scores acted as independent prognostic indicators (p < 0.001; Fig. 5D), while multivariate independent prognostic analysis showed that only age and risk scores acted as independent prognostic factors for TC patients (p < 0.001; Fig. 6E). We then incorporated risk scores with clinical factors and plotted a predictive nomogram for patients with TC (Fig. 5F). Combined with the calibration plots, it was observed that the predicted OS was consistent with the actual observations (Fig. 5G). In addition, a comparison of the c-index and the RMS of seven previously reported risk models shows that the developed model has a significant advantage (Figs. 5H, 5I). Therefore, the risk signature showed high performance in predicting OS in patients with TC in the TCGA dataset.

Figure 5 Independent prognostic factor assessment and nomogram construction.

(A–C) Area under the curve (AUC) of receiver operating characteristic (ROC) curves comparing the predictive accuracy of risk score and other prognostic factors in training, validation, and overall groups. (D, E) Univariate and multifactorial Cox analyses of clinicopathologic factors and risk score with overall survival (OS). (F) The nomogram for the prediction of 1-, 3-, and 5-year OS. (G) The calibration plots of the 1-, 3- and 5-year nomograms. (H) The comparison of C-index among risk models. (I) The comparison of restricted mean survival (RMS) among risk models.

Figure 6 Differences in tumor immune microenvironment between low and high-risk groups.

(A, B) Gene set enrichment analysis of KEGG in high-risk and low-risk groups. (C) The immune cell bubble of risk groups. (D) The box plots comparing StromalScore, ImmuneScore, and ESTIMATEScore between low and high-risk groups. (E) The single-sample gene set enrichment analysis (ssGSEA) scores of immune cells. (F) The difference of common immune checkpoint expression in the risk groups. *p < 0.05, **p < 0.01, ***p < 0.001.

Discovery of the association between tumor‑infiltrating immune cells and risk signature

We explored the underlying differences in signaling pathways between the risk groups by Gene Set Enrichment Analysis (GSEA). GSEA of the Kyoto Encyclopedia of Genes and Genomes (KEGG) pathways showed that the enriched pathways in the high-risk group involved ABC transporters, ECM-receptor interaction pathway, hedgehog signaling athway, primary bile acid biosynthesis, and tryptophan metabolism (Fig. 6A). Conversely, the accumulated pathways in the group at low risk included DNA replication, Nucleotide excision repair, Ribosome, and Spliceosome (Fig. 6B).

To elucidate the relationship between the tumor immune microenvironment (TME) and the risk signature, we performed an analysis of RNA microarray data to explore the correlation between risk characteristics and immune cells. The bar graph shows the association between risk score and tumor-infiltrating immune cells (Fig. 6C). The XCELL algorithm results showed that the risk score was significantly positively correlated with B cell plasma, plasmacytoid dendritic cells, and T cell regulatory cells (Tregs), while it was positively correlated with T cell CD4+ Th1, CD8+, and mast cells. The results of the QUANTISEQ algorithm showed that the risk score was significantly positively correlated with regulatory T cells (Tregs), while negatively correlated with macrophage M1, neutrophils, etc. The EPIC algorithm showed that the risk score was positively correlated with cancer-associated fibroblasts and negatively correlated with macrophages and NK cells. To better explore the immune status between the two groups, we evaluated the stromal score, the immune score, and the estimated score (Fig. 6D). Subsequently, we investigated the variations between the risk points and tumor infiltrating immune cells using ssGSEA. The scores for B cells, macrophages, and T helper cells were markedly higher in the group at high risk (Fig. 6E). Furthermore, we observed that there was a considerable difference (p < 0.001) between the groups in immune checkpoints such as IDO2, CD44, CD40, HHLA2, ADOARA2A, and CD80 (Fig. 6F).

Cluster analysis according to prognostic PRlncRNAs

In an effort to evaluate the immune microenvironment and response of different subtypes, cluster analysis was performed to obtain clustered subtypes. On the basis of the seven PRlncRNAs, we divided the patients into two subtypes by applying the ConsensusClusterPlus package (Fig. 7A). As shown in the Sankey diagram (Fig. 7B), most of the low-risk patients were grouped in cluster 1, whereas the majority of the high-risk patients were assigned to cluster 2. Survival analysis showed that patients in subgroup 1 had a superior OS (p < 0.001) (Fig. 7C). Based on the cluster analysis, the relationship between the different clusters and TME were examined. Box plots show that cluster 1 had higher immune, stromal, and ESTIMATE scores than cluster 2 (Fig. 7D). The heat map shows the variation in immune cell infiltration across different tumor clusters (Fig. 7E).

Figure 7 Cluster analysis.

(A) Patients were stratified into two clusters. (B) Sankey diagram; (C) K-M survival curves of overall survival (OS) in clusters. (D) Immune, stromal, and ESTIMATE scores in clusters. (E) Heatmap of immune cells in clusters.

Drug sensitivity in the PANoptosis-related LncRNA signature

We further explored the differences in drug resistance between the risk groups. We investigated the IC50 values of 80 chemotherapy drugs and inhibitors in both groups. The values of six typical drugs are shown in Fig. 8. Low-risk patients had substantially lower IC50 values for Imatinib, Sunitinib, Pazopanib, Mitomycin C, 5-Fluorouracil, and tipifarnib, suggesting that the drugs may be eligible for the treatment of patients in the low-risk group. In contrast, drugs such as Bortezomib and Phenformin showed better effects in the high-risk group. This means that candidate immunotherapeutic candidates for patients can be identified with TC based on their risk scores.

Figure 8 Drug sensitivity and correlation analysis.

(A–F) The IC50 of Imatinib, Tipifarnib, Pazopanib, Sunitinib, Bortezomib, and Phenformin in high-risk and low-risk groups. (G–L) The correlation of Imatinib, Tipifarnib, Pazopanib, Sunitinib, Bortezomib, and Phenformin with the risk score.

Assessing the expression and biological function of PRLncRNAs

Among the PRLncRNAs, we verified the gene levels of four lncRNAs in pairs of samples obtained from patients with TC in our center. Comparable expression patterns were observed in the clinical samples (Figs. 9A–9D). GAPLINC showed increased expression in tumor tissues (T) compared to in normal tissues (N), whereas RBPMS-AS1, IDI2-AS1, and LINC02154 showed lower expression levels. In addition, we also used a GEO database (GSE33630) to verify the expression differences of PRLncRNAs between thyroid cancer and normal tissues. However, we were able to obtain only the expression of RBPMS-AS1, IDI2-AS1, and GAPLINC because of limited data from the sequencing platform (Fig. 9E). As shown in Fig. S2, the RNA expression of RBPMS-AS1 was successfully upregulated in both cell lines after transfection. Functionally, its role in regulating TC cell proliferation was tested using the Cell Counting Kit-8 (CCK-8) assay (Fig. 9F). Overexpression of RBPMS-AS1 inhibited the proliferation of BHP10-3 cells significantly but did not affect TPC-1 cell growth (Fig. 9E). By wound healing assay and transwell experiments, the overexpression of RBPMS-AS1 decreased the migration ability of the two TC cell lines significantly (Figs. 9G, 9H).

Figure 9 Validation of the expression and biological function of PRlncRNAs.

(A–D) Expression analysis of GAPLINC, IDI2-AS1, LINC02154, and RBPMS-AS1 in 10 pairs of thyroid cancer tissue samples. (E) GAPLINC, IDI2-AS1, and RBPMS-AS1 expression in THCA tissues and normal tissues in GEO database. (F) The cell proliferation capacity of BHP10-3 cells after the overexpression of RBPMS-AS1. (G) Wound healing assays were used to assess the migration ability of BHP10-3 and TPC-1 cells after overexpression of RBPMS-AS1. (H) Transwell assays were used to assess the migration ability of two cell lines after overexpression of RBPMS-AS1. *p < 0.05, **p < 0.01, ***p < 0.001, and ****p < 0.0001.

Discussion

Thyroid cancer is the most prevalent malignancy of the endocrine system and is highly heterogeneous in terms of morphological features and prognosis (Bray et al., 2018). In clinical practice, the diagnosis and treatment of TC has shown promising efficacy. However, owing to the high rate of postoperative recurrence, patients still do not have a satisfactory prognosis. Thus, there is a great need to explore new biomarkers for screening high-risk groups and individual treatments. PANoptosis is a newly recognized mode of inflammatory cell death characterized by the interaction of apoptosis, pyroptosis, and necroptosis and has been shown to be present in a variety of diseases (Wang & Kanneganti, 2021). In 2020, Karki et al. (2020) found that activation of IRF1-dependent “PANoptosis” in a mouse model of colorectal cancer could prevent AOM/DSS-induced colorectal carcinogenesis. In addition, accumulating evidence suggests that a broad range of programmed cell death processes, including apoptosis, pyroptosis, and necroptosis, play an important role in TC (Holm et al., 2022). Most of the research on PANoptosis has focused on molecular mechanisms. However, few researchers have investigated the prognostic value of PRGs in tumors. Therefore, there is an urgent need to construct prognostic models based on the crucial lncRNAs of TC to guide prognostic treatment and management.

In this study, RNA-seq data of PRLncRNAs and related clinical data were obtained from TCGA database. First, we performed KEGG and GO enrichment analyses using the 35 PRGs that were mainly involved in apoptosis, pyroptosis, and necroptosis pathways. By Cox regression analyses, we identified seven lncRNAs for building a PANoptosis-related lncRNA risk model. Based on the risk scores, we stratified the TC patients into high- and low-risk groups. Survival analysis verified a statistically meaningful prognostic difference between groups (p < 0.001). Univariate and multivariate Cox regression analyses showed that the model could independently predict OS in THCA (p < 0.001). According to the ROC curves, the signature was more accurate and reliable than other clinical characteristics, and the AUC value of the test set at 1 year was 1.000. Compared with previously reported RNA or lncRNA risk models for TC, our signature is more powerful according to the c-index and RMS.

In recent years, with advancements in genomic technologies, many noncoding genes have been identified to have key roles in tumorigenesis and evolution. Among the seven PRlncRNAs, five lncRNAs (IDI2-AS1, LINC02154, C5orf34-AS1, LINC01705, and GAPLINC) were poor prognostic factors for TC, while the other two lncRNAs (DPP4-DT and RBPMS-AS1) were protective factors. Yue et al. (2022) demonstrated that LINC02154 has an imperative effect on the progression of hepatocellular carcinoma by increasing the activity of the SPC24 promoter and triggering the PI3K-AKT signaling pathway (Yue et al., 2022). According to Du et al. (2020), LINC01705 is a crucial oncogene that prevents apoptosis and accelerates proliferation of breast cancer cells (Du et al., 2020). Hu et al. (2021) revealed that DPP4 gene knockdown inhibits proliferation and epithelial-mesenchymal transition in PTC cells through suppressing the MAPK pathway (Hu et al., 2021). RBPMS-AS1 enhances radiosensitivity and apoptosis, which facilitate regulation of GBM cell proliferation (Li et al., 2022). Xu et al. (2022) reported that IDI2-AS1 is a potential prognostic marker for uveal melanoma. The results are in line with our own findings and further demonstrate their reliability.

To further explore the role of the seven PRlncRNAs in TC, we compared tumor-infiltrating immune cells between the groups. The results showed that patients in the group at high risk had a significantly higher level of immune cells, such as B cells, macrophages, and T helper cells. A previous study showed that macrophages promote tumor growth and metastasis by secreting various cytokines (Vlaicu et al., 2013). B-cell infiltration has diagnostic and prognostic value for PTC lymph node metastases (Yang et al., 2021). Meanwhile, we observed a notable decrease in the immune, stromal, and ESTIMATE scores in the high-risk group. Therefore, these results suggest that the signature has predictive capacity to reflect the immune infiltration of TC and may provide a reference for immunotherapy.

Moreover, we verified the expression of four lncRNAs (GAPLINC, IDI2-AS1, LINC02154, and RBPMS-AS1) in tumor and adjacent normal tissues, which was consistent with the expression of PRlncRNAs in the database. Of note, RBPMS-AS1, as a protective factor, was expressed less in TC. We then overexpressed RBPMS-AS1 in two types of TC cell lines and found that their migration and proliferation abilities were largely inhibited. The expression and biological function of PRlncRNAs were in line with our previous analysis.

In summary, this is the first study to use bioinformatics to construct a prognostic risk model for TC using PRlncRNAs. Our work may provide predictive tools for immune and targeted therapy for patients with TC. Although our PRlncRNA risk model achieved promising results in THCA, it still has some limitations. Our signature was established using the TCGA database only, lacking further verification of external datasets. The predictive effect of the model is yet to be validated in a large cohort of thyroid cancer patients. In addition, bioinformatics analysis is the main basis of our study, and the mechanism by which PRlncRNAs affect TC progression is uncertain. Therefore, our study provides new insights into the possible designs of PANoptosis-related drugs to treat thyroid cancer. The molecular mechanisms underlying the role of PRlncRNAs in thyroid cancer remain to be further elucidated in the future.

Conclusion

In conclusion, this analysis was the first to identify seven PANoptosis-related lncRNAs (IDI2-AS1, LINC02154, DPP4-DT, RBPMS-AS1, C5orf34-AS1, LINC01705, and GAPLINC) in TC. Furthermore, the constructed model can accurately predict OS of TC and guide targeted drugs, which holds promise for predicting the immune infiltration of tumors and guiding targeted drug therapy.

Supplemental Information

Supplemental Information 1 Sankey diagram of PANoptosis-related genes and PRlncRNAs.

Click here for additional data file.

Supplemental Information 2 The expression level of RBPMS-AS1 in BHP10-3 cells and TPC-1 cells detected by qPCR.

Click here for additional data file.

Supplemental Information 3 The cell proliferation capacity of TPC-1 cells after RBPMS-AS1 overexpression.

Click here for additional data file.

Supplemental Information 4 35 PANoptosis-related genes identified in the literature.

Click here for additional data file.

Supplemental Information 5 Clinical data from 507 TC patients.

Click here for additional data file.

Supplemental Information 6 Cell composition from the TCGA-THC A dataset samples.

Click here for additional data file.

Supplemental Information 7 The primers designed for qRT-PCR.

Click here for additional data file.

Supplemental Information 8 Code.

Click here for additional data file.

Supplemental Information 9 Raw data from the comparison of the C-index and the Restricted Mean Survival (RMS) among risk models.

Click here for additional data file.

Supplemental Information 10 Wound healing assay data.

Click here for additional data file.

Supplemental Information 11 Transwell assay data.

Click here for additional data file.

Supplemental Information 12 Raw GSE33630 sequence data.

Click here for additional data file.

Supplemental Information 13 qPCR tissue data.

Click here for additional data file.

Supplemental Information 14 qPCR data from Figure S2.

Click here for additional data file.

Supplemental Information 15 Raw cck8 data.

Click here for additional data file.

Supplemental Information 16 Sequenced RBPMS-AS1 plasmid.

Click here for additional data file.

We sincerely appreciate everyone involved in this work, as well as the public database.

Additional Information and Declarations

Competing Interests

Author Contributions

Human Ethics

Data Availability

The authors declare that they have no competing interests.

Ruowen Li conceived and designed the experiments, performed the experiments, analyzed the data, prepared figures and/or tables, authored or reviewed drafts of the article, and approved the final draft.

Mingjian Zhao analyzed the data, authored or reviewed drafts of the article, and approved the final draft.

Min Sun performed the experiments, authored or reviewed drafts of the article, and approved the final draft.

Chengxu Miao analyzed the data, prepared figures and/or tables, and approved the final draft.

Jinghui Lu conceived and designed the experiments, authored or reviewed drafts of the article, and approved the final draft.

The following information was supplied relating to ethical approvals (i.e., approving body and any reference numbers):

The Qilu Hospital of Shandong University granted Ethical approval to carry out the study within its facilities (Ethical Application Ref: KYLL-2017(KS)-248).

The following information was supplied regarding data availability:

Code and raw data are available in the Supplemental Files.

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
