# Peer review of "Construction and validation of a PANoptosis-related lncRNA signature for predicting prognosis and targeted drug response in thyroid cancer"

_PeerJ, doi:10.7717/peerj.15884_

## Round 0.1 · original submission · Major Revisions

The reviewers are all fairly positive in tone, but some additional work is required. reviewers 1 and 3 are largely stylistic and points of clarification. However, reviewer 2 makes some key points that must be addressed:

You must verify the overexpression efficiency of RBPMS-AS1 in cell lines. I regard this as essential data which should be included.

The claim that the signature found based on the informatics analysis could serve as an effective biomarker for guiding personalized treatment strategies for thyroid cancer patients requires verification to convince the field. You should consider this carefully, as without some element of verification, the study lacks impact.

Reviewer 1 ·

Basic reporting

No comment.

Experimental design

No comment.

Validity of the findings

No comment.

Additional comments

The authors proposed a PANoptosis-related lncRNA risk model and demonstrated its prognostic value through extensive computational analyses on datasets from the TCGA datasets. In general, the manuscript is well written and the analysis is sound. However, I have several minor comments for the authors to address:

1. Line 76 "https://portal.gdc. cancer.gov/": there's an extra space in the link. Also this link points to the TCGA database. I suggest providing the link to the specific gene expression data of 571 TC samples.

2. Line 86 "by incorporating the expression level": did the authors perform any normalization or preprocessing of the gene expression values?

3. Line 184-185: what was the lambda chosen for the LASSO study and what was the criterion?

4. Section 3.6 cluster analysis according to prognostic PRlncRNAs: what's the purpose of this clustering study? E.g. do the two clusters found here correspond to the real disease subtypes?

5. Figure 8: what's the hypothesis testing method used to obtain the p-values?

6. Section 3.8 assessing the expression and biological function of PRLncRNAs: is it possible to further verify the PANoptosis-related lncRNA prognostic risk model on the ten pairs of samples? The model was constructed from public datasets in TCGA. It would be a great opportunity to analyze this model on an independent dataset.

7. Line 297-298 "the AUC value of the entire set at 1 year was 0.966": suggest reporting the AUC on the test set which is more reliable.

·

Basic reporting

Li et al. constructed and validated a signature of long-coding RNAs related to PANoptosis, a form of regulated cell death, for predicting prognosis and identifying targeted drug response in thyroid cancer. The authors identified a set of lncRNAs associated with PANoptosis and used these to construct a prognostic signature.

Experimental design

1. List and summarize in more detail the data used from the TCGA and TIDE database.
2. The author did not verify the overexpression efficiency of RBPMS-AS1 in cell lines. Therefore, please supplement the effect of different overexpression efficiency of RBPMS-AS1 on cell proliferation.
3. The authors propose that the signature they found based on the informatics analysis could serve as an effective biomarker for guiding personalized treatment strategies for thyroid cancer patients is audacious, and needs more biological experiments verifications to convince the potential readers.

Validity of the findings

.

·

Basic reporting

The manuscript provides a comprehensive exploration of the role of PANoptosis-related long non-coding RNAs (PRlncRNAs) in thyroid cancer (TC), and presents an innovative prognostic model based on seven PRlncRNAs. The authors utilize a robust methodology, leveraging large-scale data from The Cancer Genome Atlas and employing sophisticated statistical analyses. Their findings not only highlight the potential of PRlncRNAs as prognostic markers but also elucidate their correlation with the tumor immune microenvironment and immune checkpoints. Additionally, the manuscript reports the potential for differential drug sensitivity in high- and low-risk TC groups. The use of in-vitro validation of lncRNAs in TC cell lines provides a significant strength to the study. Overall, this work significantly contributes to our understanding of the molecular mechanisms underlying TC, and suggests the potential of the constructed PRlncRNA prognostic model in guiding therapeutic strategies for TC patients.

Experimental design

NO comment.

Validity of the findings

No comment.

Additional comments

The author need to improve the manuscript as suggested here:

The author should tell the full name of Gene ontology (GO) and Kyoto Encyclopedia of Genes and Genomes (KEGG) pathway enrichment analyses when they mentioned these terms for the first time to make it easier for readers to understand their study. Please also pay attention to other similar situation that might confuse the readers.

Please label which images indicate the migration of cancer cells after RBPMS-AS1 overexpression in Figure 9 F.

The discussion of the results lacks critical analysis and interpretation. The authors mostly describe their findings without adequately discussing the potential implications, limitations, or future directions for research. A more comprehensive and insightful discussion would strengthen the manuscript's overall quality.

---

## Round 0.2 · accepted · Accept

Thanks for attending to the issues raised under review. I am happy to recommend acceptance now.

Reviewer 1 ·

Basic reporting

N/A

Experimental design

N/A

Validity of the findings

N/A

Additional comments

The authors have fully addressed my comments; thus I agree with the publication.

·

Basic reporting

The authors created a new model to predict thyroid cancer outcomes using specific genetic data known as PANoptosis-related long non-coding RNAs (PRlncRNAs). They sourced this data from a well-respected database called The Cancer Genome Atlas (TCGA). The model can assign a risk score that reveals information about the tumor's environment, the body's immune response, and possible medication responses. Furthermore, using quantitative real-time polymerase chain reaction (qRT-PCR), they've found that four specific PRlncRNAs show different levels in cancerous and healthy tissues. This breakthrough model not only broadens our understanding of thyroid cancer but also helps doctors choose the most effective treatment options.

In response to reviewers' questions, they also examined the expression of RBPMS-AS1 in thyroid cancer tissues and cell lines.

Experimental design

The experiments were properly designed.

Validity of the findings

The findings are convincing.

Additional comments

None.